# Improving the Efficiency of Geographic Target Regions for Healthcare Interventions

**DOI:** 10.3390/ijerph192214721

**Published:** 2022-11-09

**Authors:** Matthew Tuson, Matthew Yap, Mei Ruu Kok, Bryan Boruff, Kevin Murray, Alistair Vickery, Berwin A. Turlach, David Whyatt

**Affiliations:** 1Department of Mathematics and Statistics, The University of Western Australia, Crawley, WA 6009, Australia; 2Medical School, The University of Western Australia, Crawley, WA 6009, Australia; 3UWA School of Agriculture and Environment, The University of Western Australia, Crawley, WA 6009, Australia; 4Department of Geography, The University of Western Australia, Crawley, WA 6009, Australia; 5School of Population and Global Health, The University of Western Australia, Crawley, WA 6009, Australia

**Keywords:** geographic target regions, healthcare interventions, targeting efficiency, logistical factors, Marxan, MinPatch, Spatial Targeting Algorithm

## Abstract

Appropriate prioritisation of geographic target regions (TRs) for healthcare interventions is critical to ensure the efficient distribution of finite healthcare resources. In delineating TRs, both ‘targeting efficiency’, i.e., the return on intervention investment, and logistical factors, e.g., the number of TRs, are important. However, existing approaches to delineate TRs disproportionately prioritise targeting efficiency. To address this, we explored the utility of a method found within conservation planning: the software Marxan and an extension, MinPatch (‘Marxan + MinPatch’), with comparison to a new method we introduce: the Spatial Targeting Algorithm (STA). Using both simulated and real-world data, we demonstrate superior performance of the STA over Marxan + MinPatch, both with respect to targeting efficiency and with respect to adequate consideration of logistical factors. For example, by design, and unlike Marxan + MinPatch, the STA allows for user-specification of a desired number of TRs. More broadly, we find that, while Marxan + MinPatch *does* consider logistical factors, it also suffers from several limitations, including, but not limited to, the requirement to apply two separate software tools, which is burdensome. Given these results, we suggest that the STA could reasonably be applied to help prevent inefficiencies arising due to targeting of interventions using currently available approaches.

## 1. Introduction

Worldwide, health authorities are tasked with efficiently distributing finite healthcare resources to address disease among populations. Often, this requires appropriate prioritisation of geographic target regions (TRs) for intervention. Such prioritisation has been widely proposed or realised previously, including to address both infectious and non-infectious diseases [1,2,3,4]. For example, spatial prioritisation of resources has recently been suggested to guide distribution of COVID-19 vaccines [5,6].

In delineating TRs, both ‘targeting efficiency’, i.e., the return on intervention investment, and logistical factors, e.g., the number of TRs, are important [7]. In particular, it is important to consider both the number of TRs and their size and degree of compactness. This is because it might be infeasible for authorities to intervene in a large number of locations, and certain interventions might be best located within regions of certain sizes or shapes. Testing clinics, for example, might be best located within relatively large, compact TRs, in order to efficiently service correspondingly large population sizes while limiting travel time to and from the new clinics for staff and patients. However, recent approaches to delineate TRs have disproportionately prioritised targeting efficiency. For example, Lessler et al. (2018) [3] described how the efficiency of oral cholera vaccine distribution in sub-Saharan Africa could be maximised through prioritising distribution to 20 × 20 km grid cells with the highest rates of infection [3]. Similarly, Coburn et al. (2017) [8] described how the efficiency of HIV interventions in Lesotho could be maximised through targeting 1 × 1 km grid cells with the highest density of infection [8]. However, it would likely be infeasible to target HIV interventions to the numerous, discontiguous TRs delineated by Coburn et al. (2017) [8]. For this and other reasons, Lessler et al. (2018) [3], rather than advocating the optimally efficient strategy described above, instead suggested prioritising oral cholera vaccine distribution in Africa by either: (1) local administrative districts (continent-wide), or (2) countries, and subsequently districts within each country. However, both of these approaches are undermined by the modifiable areal unit problem (MAUP) [9] due to relying on the district boundaries. Briefly, the MAUP describes how results that are based on areal units will depend on the unit boundaries. In the present context, this means that, in general, pre-defined, ‘single-aggregation’ administrative boundaries will not adequately represent the geographic distribution of a given disease, except possibly by chance. Recognising this, Tuson et al. (2020) [7] showed how the MAUP can be mitigated through targeting interventions guided by smoothed maps of fine-resolution data. However, that approach did not allow for adequate control over either the number of TRs or their size and degree of compactness.

Given these limitations, we endeavored to search outside of health for tools that could address these issues. In particular, we were interested in tools that sought to optimise targeting efficiency while also considering logistical factors, and, with regards to the latter, in particular: (1) the number of TRs, and (2) the size and degree of compactness of individual TRs within a given set. However, only one such tool was forthcoming: a method found within conservation planning, namely the software Marxan [10] and an extension, MinPatch [11] (hereafter ‘Marxan + MinPatch’). Briefly, Marxan + MinPatch can be used to produce ‘portfolios’ (i.e., sets) of ‘protected areas’ (PAs, or TRs), which in turn may be used to guide conservation of one or more ‘abundance features’ while minimising associated costs. Marxan was developed to guide systematic, multi-objective planning in conservation, while MinPatch can be applied to sets of Marxan portfolios to ensure a user-specified minimum size for individual PAs within a given portfolio. Abundance features examined using Marxan are typically species of plants or animals, and the cost of conservation is generally the economic cost of purchasing PAs. However, in the context of health, abundance features examined using Marxan + MinPatch might be cases of disease or other health events, such as hospital admissions, and the cost of conservation might be the population size or geographic area of regions to be conserved (i.e., targeted). For example, to help control the spread of infectious diseases such as COVID-19, Marxan + MinPatch could be applied to delineate sets of TRs for testing or vaccine distribution that collectively contain a minimum proportion of infected individuals while minimising the target population size/area.

Unfortunately, application of Marxan + MinPatch suffers from several limitations. First, the requirement to apply two separate software tools, which is potentially burdensome. Second, the incorporation within both Marxan and MinPatch of a global, rather than a local, parameter (the boundary length modifier (BLM)) to control for the degree of fragmentation among TRs in their output portfolios [11]. Third, MinPatch’s reliance on a distance-based radius when defining new TRs, which limits flexibility when examining irregularly shaped units such as administrative boundaries. And fourth, an inherent potential of MinPatch to output sub-optimally targeting efficient portfolios due to the ‘top-down’ nature of its algorithm. Regarding the latter, within MinPatch, the locations of new TRs are defined *before* those TRs are whittled to reduce cost. Therefore, it is likely that a ‘bottom-up’ approach, where TRs are purpose-built from the ground up, would yield greater targeting efficiency. Together, these limitations suggest that superior approaches might be developed.

Accordingly, in this paper, we compare the utility of Marxan + MinPatch for efficiently delineating geographic TRs for healthcare interventions, to that of a new method we introduce: the Spatial Targeting Algorithm (STA). By design, the STA incorporates comparatively greater control over logistical features, for example through allowing for user-specification of a desired number of TRs. However, the relative targeting efficiency of the two algorithms in the context of health was unknown. Therefore, we compared the two methods using both simulated and real-world data, the latter a dataset of ischaemic stroke hospital admissions in Perth, Western Australia (WA) that has been previously examined in the literature [7]. As noted by Tuson et al. (2020) [7], strokes require rapid intervention to avoid irreparable damage to nerve tissue [12], which necessitates ongoing consideration of patients’ access to essential stroke services, such as specialist hospital units and ambulance depots. This endeavour is supported by the precise, efficient and logistically effective delineation of TRs for the placement of such services, a practice which both Marxan + MinPatch and the STA inform.

## 2. Materials and Methods

### 2.1. Marxan + MinPatch

To apply Marxan, users specify: (1) an input set of ‘planning units’; (2) the number of independent ‘runs’ to be undertaken; and (3) a BLM to be applied in each run. The number of runs corresponds to the number of output portfolios, while the BLM governs the degree of fragmentation amongst PAs within a given portfolio; relatively high BLMs result in relatively little fragmentation, and vice versa. For a given set of portfolios, Marxan automatically designates as ‘optimal’ the portfolio that minimises cost; however, users may consider one or more portfolios when making decisions.

Applied to a set of Marxan portfolios, for each portfolio, MinPatch: (1) removes PAs that are smaller than the specified minimum size; (2) adds new PAs according to a specified radius; and (3) applies ‘simulated whittling’ to both the new and original PAs in order to reduce cost. Here, simulated whittling describes an iterative process of identifying and removing planning units located on the boundaries of PAs, while maintaining the specified abundance targets and enforcing the minimum size. MinPatch also incorporates a BLM that is often, but not necessarily always, the same as that specified for Marxan [11]. Depending on the type of output desired, one or more of MinPatch’s steps may be excluded; for example, simulated whittling might be excluded if portfolios of relatively compact PAs are desired. MinPatch produces one portfolio for each of a given set of Marxan portfolios. Similarly to Marxan, it designates as ‘optimal’ the portfolio that minimises cost; however, again, users may consider one or more portfolios when making decisions.

### 2.2. The Spatial Targeting Algorithm (STA)

The STA involves creating numerous, differently shaped polygons at a user-specified scale, and ‘targeting’ these polygons to delineate sets of TRs for intervention. Specifically, it comprises the following steps:


**Step 1. Specify a set of ‘minimal-resolution’ spatial units (hereafter ‘minimal units’)**


The minimal units are analogous to the planning units specified for Marxan. Usually, they will be a set of fine-resolution administrative units or grid cells. In Australia, for example, they might be Australian Bureau of Statistics (ABS) Statistical Areas Level 1 (SA1s), for which minimally aggregated health and Australian Census population counts are obtainable. Comparable units in other countries include Census Blocks in the US and Output Areas in the UK, though the latter are generally smaller than SA1s. To apply the STA, users must specify:


**Step 2. Specify parameters**


The number of polygons to create (specified per minimal unit);A target polygon size (typically a population size or geographic area);One or more weighting functions; andA target percentage of cases (or other health events, e.g., hospital admissions).

The number of polygons to create should be chosen based on a rule of thumb developed later; briefly, it will be suggested that the number of polygons to create should be chosen such that the algorithm’s output is stable while maintaining computational feasibility. Meanwhile, the target polygon size should be chosen guided by the characteristics of a proposed intervention; for example, if an intervention is planned that will target relatively large TRs, then a relatively large target polygon size will be appropriate. The weighting functions govern the general shape of the polygons created and thence the TRs delineated. For example, specification of an inverse distance-based weighting function will lead to relatively compact polygons being created, and consequently to correspondingly compact TRs being delineated. By contrast, specification of a number of cases-based weighting function will lead to relatively ‘tree-like’ polygons being created, and consequently to correspondingly tree-like TRs being delineated. As for the target polygon size, the choice of weighting functions should be guided by the characteristics of a planned intervention. For example, if an intervention is planned that will be targeted to relatively compact regions (as will often be the case in health), then specification of an inverse distance-based weighted function might be appropriate. Finally, the target case percentage should reflect corresponding targets specified for planned interventions; for example, a target case percentage of 50% might be specified to align with a corresponding target specified for a treatment intervention aimed at limiting the spread of an infectious disease.


**Step 3. Create polygons**


In this step, the STA designates each minimal unit in turn as the ‘seed’ unit and creates the specified number of polygons beginning with that unit. For each polygon, this involves iterating the following steps:Identify minimal units neighbouring either:a.The seed unit, if iteration = 1, orb.Any unit already selected, if iteration > 1;Compute a ‘selection probability’ for each neighbouring unit; andSelect one neighbouring unit through sampling from a multinomial probability distribution defined by the set of computed selection probabilities. If the target polygon size is reached, end.

In step ii, each of the specified set of weighting functions $w_{i}$, i=1,…,I are evaluated for each neighbouring unit u to compute sets of weights $w_{i}\left(u\right)$. Here, for simplicity, we suppress in our notation the dependency of the weights on the seed unit. For each weighting function in turn, the resulting weights are then normalised across units u:(1)$$w_{norm,\;i}\left(u\right)=\frac{w_{i}\left(u\right)}{\sum_{u}w_{i}\left(u\right)}$$

The product of the normalised weights is taken across weighting functions, for each unit u:(2)$$w_{u}=\underset{i}{\prod }w_{norm,i}\left(u\right)$$
and the resulting products are again normalised across units u to compute a set of selection probabilities:(3)$$w_{norm,u}=\frac{w_{u}}{\sum_{u}w_{u}}$$
where $w_{norm,u}$ is the selection probability computed for unit u.


**Step 4. Delineate target regions**


To delineate a set of TRs, the STA iteratively targets the set of created polygons until the set of targeted polygons collectively contains at least the pre-specified target percentage of cases. Specifically, the following steps are iterated:Order available polygons by their values, from highest to lowest; thenTarget (i.e., remove) the polygon with the highest value. If the target percentage of cases is reached, end; otherwise, exclude overlap with the targeted polygon from all remaining polygons and recalculate the values of any affected polygons.

The value calculated for each polygon will typically be a rate or a density, e.g., cases per unit capita or geographic area. Furthermore, it will usually be related to the specified target polygon size; for example, if the chosen value is cases per unit capita, then the target polygon size might be population size. Figure 1 visually illustrates the architecture of the STA in a flow chart.

### 2.3. Simulation Study

In order to: (1) illustrate the STA’s application, and (2) compare the two methods’ performance, a simulated point location dataset of 100 disease cases was generated by applying a multinomial probability distribution across a square, spatially correlated random field. The random field was created using functions within the *gstat* package in R version 4.0.3 [13], following an online guide (http://santiago.begueria.es/2010/10/generating-spatially-correlated-random-fields-with-r/, accessed on 24 August 2021), while the generation of cases was undertaken primarily using the *rmultinom()* function in base R and the *spsample()* function in the *sp* package, among other functions. The values used in each of these functions were chosen in order to generate a sensible dataset.

#### 2.3.1. Illustration of the STA’s Application

To illustrate the STA’s application, for simplicity, we defined a 5 × 5 unit grid overlaying the simulated field to be the set of minimal units specified for the STA, and we specified:that only 1 polygon be created per minimal unit;a target polygon size of 5 minimal units;no weighting functions;‘cases per minimal unit’ to be the value calculated for each polygon; anda target case percentage of 40% of cases.The STA was applied using R.

#### 2.3.2. Comparison of the STA to Marxan + MinPatch

To compare the two methods’ performance, we defined a 20 × 20 unit grid overlaying the simulated field to be the set of minimal units specified for the STA and the set of planning units specified for Marxan + MinPatch. Marxan + MinPatch was applied via the Conservation Land-Use Zoning software (CLUZ) plug-in to QGIS [14]. To ensure a fair comparison, and to highlight each method’s flexibility, we specified:a minimum size for each PA within MinPatch and a corresponding target polygon size within the STA of 16 minimal units;a target abundance proportion within Marxan and a corresponding target case percentage within the STA of 50% of cases;‘cases per minimal unit’ to be the value calculated for each polygon within the STA, and the related ‘cost of conservation’ within Marxan to be 1 per planning unit;BLMs of 0, 0.001, 0.002 and 0.005 within Marxan + MinPatch (preliminary analyses gave no indication that doing so would usefully extend the simulation, so we did not vary the BLM specified between Marxan and MinPatch);radii of 2.25, 3 and 5, and either simulated whittling or no simulated whittling within MinPatch; andan inverse distance- or value-based weighting function, and either splitting or no splitting within the STA.

Finally, and although attainment of complete equivalence in this aspect was impossible, we specified that: 10 polygons be created per minimal unit within the STA, and 10 runs be undertaken within Marxan. Thus, 24 sets of 10 MinPatch portfolios and four STA portfolios were produced in total, the former corresponding to the different combinations of BLM, radius and application (or not) of simulated whittling (4 values × 3 values × 2 options = 12), and the latter corresponding to the different combinations of weighting functions and the different options for splitting (2 values × 2 values = 4). However, to facilitate a fair comparison, we considered only the portfolios designated as ‘optimal’ for the 24 sets of MinPatch portfolios (i.e., one per set). We refer to the STAs applied with the inverse distance- and value-based weighting functions as the ‘distance-weighted’ and ‘value-weighted’ STAs, respectively. For the distance-weighted STA, weights for neighbouring units u in the polygon creation process were calculated as
(4)$$w\left(u\right)=\frac{1}{d_{u}{}^{p_{d}}}$$
where $d_{u}$ is the Euclidean distance between the geographic centroids of unit u and the seed unit and $p_{d}$ is a weighting factor, which we set equal to 25 to ensure that only compact TRs are delineated. For the ‘value-weighted’ STA, corresponding weights for neighbouring units u were calculated as:(5)$$w\left(u\right)=v_{u}{}^{p_{v}}$$
where $v_{u}$ is the value of unit u and $p_{v}$ is a weighting factor, which we again set equal to 25 to ensure that only tree-like TRs are delineated.

Two comparisons were undertaken: (1) of the portfolios produced using the two methods that contained maximally compact TRs, and (2) of the corresponding portfolios that contained maximally targeting efficient TRs. Portfolios produced using the distance-weighted STA applied without splitting and Marxan + MinPatch applied without simulated whittling were considered to be candidates for designation as maximally compact, while all portfolios produced using the two methods were considered to be candidates for designation as maximally targeting efficient. To avoid the undue impact of idiosyncrasies of any particular dataset, the comparisons were repeated for an additional nine datasets that were simulated in the same manner as the one described previously. More datasets than these were not considered due to the inhibitory point-and-click nature of Marxan + MinPatch’s application within CLUZ. Thus, based on the ten datasets, for each comparison we report: (1) the mean difference in targeting efficiency between the portfolios produced using the STA and Marxan + MinPatch, and (2) the range of observed differences, across the ten datasets.

### 2.4. Comparison of the STA to Marxan + MinPatch

Following Tuson et al. (2020) [7], we defined the study area of Perth to comprise the five Greater Perth ABS Statistical Areas Level 4 (SA4s): “Perth—Inner”, “Perth—South East”, “Perth—South West”, “Perth—North East”, and “Perth—North West”, excluding two single-SA1 islands: Rottnest Island and Garden Island, due to the primary purpose of those islands being to operate as tourist/day-trip destinations and to house the Australian Navy’s largest fleet base, respectively. We defined SA1s to be the set of minimal units specified for the STA and the set of planning units specified for Marxan + MinPatch, and we obtained 2016 Australian Census population data for Perth, stratified by SA1, via the ABS’ web-based TableBuilder tool.

The stroke dataset consists of admissions by Perth residents to WA hospitals with principal International Classification of Diseases, Tenth Revision, Australian Modification) [15] diagnoses of I63 (Cerebral infarction), I64 (Stroke, not specified as haemorrhage or infarction) or H34.1 (Central retinal artery occlusion). We extracted these data from the WA Hospital Morbidity Data Collection (HMDC). Following various exclusions (see Tuson et al., 2020 [7]), 2523 admissions were available for analysis; these we aggregated by SA1.

It has previously been suggested that targeting interventions such as mobile stroke units to address stroke in Perth might be guided by portfolios of relatively large, compact TRs [7]. Therefore, we applied the distance-weighted STA without splitting and Marxan + MinPatch without simulated whittling. Within the STA, we specified the same weighting function as for the distance-weighted STA in the simulation study. Further, to ensure a fair comparison, we specified:A target polygon size within the STA and a corresponding minimum size for each PA within MinPatch of 11,250 people; andThe crude rate of stroke admissions to be the value calculated for each polygon within the STA, and population size to be the related cost of conservation within Marxan + MinPatch.

Furthermore, we specified an arbitrary target of 15% of admissions for both methods. This value is the same as that used previously by Tuson et al. (2020) [7], while the target polygon size of 11,250 was chosen to approximately match the mean population size of SA2s in 2016 (11,248; [7]). We tested various BLMs within Marxan + MinPatch; however, observing little difference in the resulting portfolios, we (arbitrarily) selected zero. Within MinPatch, we specified a radius of 0.011; this was the smallest value that could be defined in order to delineate maximally compact TRs while still maintaining the specified target admissions percentage and satisfying the minimum size constraint. Finally, based on a rule-of-thumb suggested for the STA in the simulation study, we specified that 10 polygons be created per minimal unit within the STA and that 10 runs be undertaken within Marxan + MinPatch.

## 3. Results

### 3.1. Illustration of the STA’s Application to Simulated Data

This section describes the application of the STA to the simulated point location dataset described in Methods. Figure 2 shows the 25 polygons created in the STA’s polygon creation step (‘Step 3’ in Section 2.2). Since no weighting functions were specified, each polygon’s shape was determined randomly. The polygons’ values ranged between 1 and 7.4 cases per minimal unit, with both polygon 13 and polygon 20 having the maximum value. For the full range of values of polygons in Figure 2, see Appendix A. Thus, in the STA’s targeting step (‘Step 4’ above), either of those polygons could have been targeted first. The algorithm arbitrarily selected and removed polygon 13, and subsequently removed overlap with that polygon from all remaining polygons; Figure 3 shows the result. Since polygon 13 contained 37 cases, i.e., 37% of all cases, which is less than the specified target of 40% of cases, the algorithm continued.

### 3.2. Splitting

For some polygons in Figure 2, removal of overlap with polygon 13 resulted in multiple, discontiguous polygon fragments being defined (e.g., see ‘polygon’ 15 in Figure 3). In subsequent iterations, these fragments could either be considered to be independent or not. If the former, we refer to the STA as having been applied “with splitting”, and if the latter, “without splitting”. In the case of polygon 15, supposing the STA was applied with splitting in this case, two polygon fragments remained; these comprised two minimal units each and had values of 5 and 1 case(s) per minimal unit, respectively. The values of all polygons (and polygon fragments) represented in Figure 3 ranged between 0.667 and 8 cases per minimal unit, with only polygon fragment 7B (see Figure 3) having the maximum value. For the full range of values among polygons and polygon fragments in Figure 3, see Appendix A. Thus, that fragment was targeted second (i.e., in iteration 2 of the targeting algorithm). Since it contained eight cases, 45 cases (and thus 45% of all cases) had then been targeted in total and the algorithm ended. Figure 4a shows the resulting TRs—only one TR, in fact—the targeting efficiency of which can be expressed as 7.5 cases per minimal unit (calculation: 45 cases/6 minimal units).

Supposing the STA was instead applied without splitting in the above case, the values of the 25 ‘polygons’ represented in Figure 3 ranged between 0.667 and 7.5 cases per minimal unit, with only polygon 20 having the maximum value. Thus, that polygon was targeted second. Since it contained 30 cases, 67 cases (and 67% of all cases) had then been targeted in total, and again the algorithm ended. Figure 4b shows the resulting TRs—again only one TR, in fact—the targeting efficiency of which can be expressed as 7.444 cases per minimal unit (calculation: 67 cases/9 minimal units). This value is slightly lower than that of the single TR delineated using the STA applied with splitting (7.5 cases per minimal unit), reflecting the fact that, in general, the targeting efficiency of portfolios produced using the STA applied with splitting will exceed that of portfolios produced using the STA applied without splitting due to the increased degree of flexibility that splitting allows.

It is important to note that neither of the portfolios shown in Figure 4 are particularly (targeting) efficient. This is because only a simple application of the STA was undertaken in order to convey the method; no weighting functions were specified, and only one polygon was produced per minimal unit. Further, in both cases, the percentage of cases ultimately targeted using the STA exceeded the specified target of 40% of cases. This is because the STA’s targeting step, by design, identifies the set of maximally efficient TRs that contains at least the specified target percentage of cases. In this characteristic, the STA is similar in nature to Marxan + MinPatch, the application of which may result in somewhat larger percentages than the specified targets of one or more abundance features being earmarked for conservation. However, the relatively large differences between the target and targeted percentages observed here simply reflect the rudimentary nature of the simulation.

### 3.3. Comparison of the STA to Marxan + MinPatch Using Simulated Data

Figure 5 depicts the locations of the simulated dataset and the 20 × 20 unit grid of planning units specified for Marxan. Additionally shown are the optimal portfolios designated for the four sets of 10 Marxan portfolios (i.e., those based on the BLMs of 0, 0.001, 0.002 and 0.005; Figure 6a–d, respectively). These portfolios evidence the inverse relationship that exists between the BLM specified for Marxan and the degree of fragmentation among TRs in its output portfolios [11]. Additionally demonstrated is the known shortcoming of Marxan that varying the BLM does not provide for sufficient control over the size and shape of individual TRs within a given portfolio [11]. For example, in Figure 6d, while one TR is large and relatively elongated, the others are small and relatively compact.

Figure 6 shows the optimal portfolios designated for the 12 sets of MinPatch portfolios produced through applying MinPatch without simulated whittling to the four sets of Marxan portfolios, and Figure 7 shows the corresponding portfolios designated through applying MinPatch with simulated whittling. In these figures, two results are demonstrated: first, and most strikingly, many of the portfolios in each figure are identical; specifically, in both figures, the portfolios based on BLMs of 0, 0.001 and 0.002 are identical for a given radius (Figure 6a,d,g and Figure 7a,d,g). This is because the TRs in the underlying Marxan portfolios (Figure 5) were smaller than the minimum size of 16 minimal units specified for MinPatch. Thus, within MinPatch, those TRs were removed and, regardless of the specified BLM, replaced by the same set of new, larger TRs that were subsequently either whittled or not. By contrast, the Marxan TRs that were based on a BLM of 0.005, which sometimes satisfied the minimum size constraint (e.g., see the large TR in Figure 5d), being not removed by MinPatch, are consequently either fully (e.g., in the case of Figure 6) or partially (e.g., in the case of Figure 7) represented in the final MinPatch portfolios.

The second result demonstrated is that, unexpectedly, the portfolios shown in Figure 7, despite comprising TRs that are relatively tree-like, are not maximally efficient. For example, the single TR in Figure 7a could have been further whittled while still maintaining the target percentage of cases and the specified minimum size constraint. Since application of simulated whittling within MinPatch purportedly leads to delineation of maximally efficient TRs [11], this result bears further investigation.

Table 1 shows targeting efficiency and number of TRs values for each of the optimal MinPatch portfolios (Figure 6 and Figure 7). These data demonstrate that, as expected, regardless of the specified BLM and radius, portfolios produced using Marxan + MinPatch applied with as opposed to without simulated whittling exhibit greater targeting efficiency. For example, the efficiency of the optimal MinPatch portfolio that was based on a BLM of zero, a radius of 2.25 and simulated whittling was 1.162 cases per minimal unit, while the efficiency of the corresponding portfolio produced through applying MinPatch without simulated whittling was 0.847 cases per minimal unit. None of the 24 portfolios in Figure 6 and Figure 7 contained more than two TRs.

Figure 8 shows the portfolios produced through applying the distance- and value-weighted STAs with and without splitting (the distance-weighted STA: Figure 8a,b, respectively; the value-weighted STA: Figure 8c,d, respectively). As expected, the TRs delineated using the distance-weighted STA are relatively compact, while the TRs delineated using the value-weighted STA are relatively tree-like. Targeting efficiency and number of TRs values for the four STA portfolios are shown in Table 2. For example, for the distance-weighted STA applied with splitting (Figure 8a), to reach the specified target of 50% of cases, targeting of three discontiguous regions comprising 54 minimal units (i.e., 13.5% of all units) was required (Table 2). By comparison, for the value-weighted STA, also applied with splitting (Figure 8c), to reach 50% of cases, targeting of two discontiguous regions comprising 32 minimal units (i.e., 8% of all units) was required (Table 2). These results demonstrate the impact of imposing a progressively more stringent compactness constraint within the STA: correspondingly reduced targeting efficiency.

The STA portfolio of maximally compact TRs was that produced using the distance-weighted STA applied without splitting (Figure 8b), while the corresponding Marxan + MinPatch portfolio was that produced based on a BLM of 0.005, a radius of 2.25 and no simulated whittling (Figure 6j). Between these portfolios, the STA portfolio had one additional TR (three TRs as compared to two) but greater targeting efficiency (1.014 versus 0.98 cases per minimal unit). Across all ten datasets, the mean percentage increase in efficiency attained through applying the STA as opposed to Marxan + MinPatch was 54.8% (range: 14.8–95.5%).

The STA portfolios of maximally targeting efficiency were those produced using the value-weighted STA applied either with or without splitting (Figure 8c,d), while the corresponding Marxan + MinPatch portfolios were those produced based on BLMs of 0, 0.001 or 0.002, radii of either 2.25 or 5 and simulated whittling (Figure 7a,c,d,f,g,i). Between these portfolios, the STA portfolios again had greater targeting efficiency (1.563 versus 1.163 cases per minimal unit), while the number of TRs varied. Across all ten datasets, the mean percentage increase in efficiency associated with applying the STA as opposed to Marxan + MinPatch was 17.2% (range: 2.0–34.4%).

### 3.4. Extended STA Results

The following sections describe certain extended results of the STA.

#### 3.4.1. Sensitivity to the Number of Polygons Created Per Minimal Unit within the STA

Theoretically, the creation of each new polygon within the STA will increase targeting efficiency until the maximum possible efficiency is reached. However, this increase will be offset by a corresponding increase in computation time. Therefore, as a rule of thumb, when applying the STA, we suggest that additional polygons be created until the resulting increase in efficiency is marginal. To illustrate, in Appendix A we show how the targeting efficiency of the distance- and value-weighted STAs, applied to 100 datasets simulated in the same manner as those described previously, increases only marginally, on average, when creating more than approximately 15 and 10 polygons per minimal unit, respectively.

#### 3.4.2. Exact Specification of a Desired Number of TRs

To this point, the number of TRs in portfolios produced using both the STA and Marxan + MinPatch has been outside of the users’ control. In practice, however, specification of this number might be desired, for example when funding was available to build a specific number of new community health clinics. Therefore, below, we describe how, in such cases, a variation of the STA can be applied to obtain appropriate solutions.

To illustrate, suppose that, for the simulated dataset described previously, it is desired to delineate the maximally efficient set of two compact, non-overlapping TRs, each of size 16 minimal units. Theoretically, provided enough polygons were created, such a set could be identified from among all possible pairs of polygons underlying the distance-weighted STA, applied without splitting. However, if many polygons were created, it might be computationally infeasible to identify that pair. Therefore, a greedy algorithm [16] might be applied: select the polygon with the highest value, then the one with the next highest value which does not overlap the first, and so on. However, this approach might identify a sub-optimally efficient solution. Therefore, to avoid this, we propose the following steps: order available polygons by their values, from highest to lowest, then iterate:Discard a constant, pre-specified percentage of polygons, beginning with those ranked lowest (i.e., those with the lowest values);Select each remaining polygon in turn to form a new branch of the targeting algorithm; andIn each unique branch, discard any of the remaining polygons that overlap the one selected. If the specified number of polygons is reached in all branches, or there are no remaining polygons, end.

The above algorithm outputs a reduced number of polygon sets—pairs, in the example above—from among which the one that is maximally efficient can more feasibly be selected. Specifying a discard percentage of 80%, we applied the algorithm to the problem posed above; Figure 9 shows the result. The two TRs shown together comprise 32 minimal units and 39 cases; thus, their targeting efficiency can be expressed as 1.22 cases per minimal unit (calculation: 39 cases/32 minimal units).

To investigate the impact of choosing different discard percentages, we re-applied the algorithm to the same simulated dataset while varying the discard percentage between 0% and 95%. Interestingly, the same set of TRs was produced each time. Therefore, as a rule of thumb, and in order to manage computation time, we suggest that the algorithm be applied with a relatively high discard percentage, e.g., 80%, as we have done above.

#### 3.4.3. Application of the STA to Modelled, or Smoothed, Risk Surfaces

In many situations, only modelled, or smoothed, risk surfaces are available to guide intervention targeting. Examples include the maps of cholera and HIV risk found within Lessler et al. (2018) [3] and Coburn et al. (2017) [8], respectively, cited previously, and maps of malaria risk found within Weiss et al. (2019) [17]. The production and display of such maps is ubiquitous both in the literature and in practice. Application of the STA in such cases proceeds as described above, except that the set of minimal units is automatically defined by the geographic resolution of the surface. An illustration of this idea is provided in Appendix A, where we apply the STA to smoothed risk surfaces created using a recently proposed smoothing technique—the Overlay Aggregation Method (OAM; [7]).

### 3.5. Real-World Application: Stroke

Figure 10a maps the 4248 SA1s and 164 SA2s that defined the geography of Perth in 2016. Figure 10b maps Perth’s 2016 population density by SA2. These maps reproduce Figure 6 in Tuson et al. (2020) [7]. Perth’s population, which in 2016 was 1.85 million, straddles the Swan and Canning Rivers, and sprawls north-to-south along the coastline.

Figure 11a,b map the sets of TRs delineated for stroke using Marxan + MinPatch and the STA, respectively. As expected, given (1) the inverse distance-based weighting function specified for the STA, and (2) that no simulated whittling was applied within MinPatch, the TRs delineated using both methods were relatively compact (while conforming to the somewhat variable shapes of the SA1 boundaries). For Marxan + MinPatch, eight TRs were delineated in total; these contained 15.02% of strokes and 7.58% of the population; thus, their targeting efficiency can be expressed as 1.98% of strokes for every 1% of the population targeted (calculation: 15.02/7.58). By comparison, for the STA, 10 TRs were delineated in total; these contained 16.33% of strokes and 6.59% of the population; thus, their targeting efficiency can be expressed as 2.48% of strokes for every 1% of the population targeted (calculation: 16.33/6.59). These results are consistent with those of the simulation study: the targeting efficiency of the STA was greater than that of Marxan + MinPatch, while the number of TRs was similar.

## 4. Discussion

In this paper, we have described a limitation of currently available methods for delineating sets of geographic TRs for healthcare interventions, namely the disproportionate prioritisation by those methods of ‘targeting efficiency’, i.e., the return on intervention investment, over important logistical factors such as the number of TRs, and aimed to address this limitation through comparing the utility of a method found within conservation, Marxan + MinPatch, to that of a new method we introduce, the STA. This comparison showed favourable performance of the STA over Marxan + MinPatch, both with respect to targeting efficiency and with respect to adequate consideration of logistical factors, the latter primarily by design. These findings suggest that the STA might usefully be applied to guide geographically targeted resource allocation and other interventions in a range of health contexts.

The superior performance of the STA over Marxan + MinPatch observed in this study can be attributed to the limitations of the latter noted in the Introduction, namely: (1) the incorporation within both Marxan and MinPatch of a global, rather than a local parameter (the BLM) to control for the degree of fragmentation among TRs in their output portfolios; (2) MinPatch’s reliance on a distance-based radius when defining new TRs; and (3) the inherent potential of MinPatch to output sub-optimally targeting efficient portfolios due to the ‘top-down’ nature of its algorithm. Regarding (1), it is clear that a global parameter cannot be expected to optimise either the targeting efficiency or orientation/configuration of individual TRs within a given portfolio. Similarly, regarding (2), it is not unexpected that, in health-related research, where the geographic units being examined are usually pre-defined and irregularly shaped, a distance-based radius would underachieve when compared to one that is flexible in terms of the shapes it is able to build. And finally, regarding (3), it is again not unexpected that an algorithm which builds flexible shapes from the ground up would yield greater efficiency than one which builds ‘top-down’. Nevertheless, it is worth reiterating the additional, unexpected result of this study: that some of the TRs output by MinPatch were not maximally efficient, despite being delineated using whittling. As we have written, this result bears further investigation.

In line with previous suggestions [7], the set of ten, compact TRs delineated using the STA for stroke in Perth might represent suitable targets for interventions such as mobile stroke units. However, it is outside of the scope of this paper to provide, in a given scenario, detailed guidance regarding either: (1) what interventions might be appropriate, or (2) how such interventions should be implemented. Rather, we simply provide the STA as a flexible tool that can effectively be used to delineate sets of geographic TRs for whatever intervention is deemed potentially most useful by planners. However, to demonstrate the STA’s flexibility, we note that, in the context of stroke, weighting functions based on distance to existing ambulance depots or stroke units might be specified in place of, or in addition to, the Euclidean distance-based weighting function we have used here. Furthermore, different choices of admissions targets might be used (here, following Tuson et al., 2020 [7], we used 15%), and the sensitivity of the results to that value might be investigated. Such an investigation represents an interesting avenue for future work.

In the context of health, our focus on efficiency as a means of comparing the output of the STA to that of Marxan + MinPatch is relatively novel; while we have cited several papers that explicitly evaluate targeting efficiency, such papers are, in fact, relatively rare. Instead, studies examining the geographic distribution of disease and other health outcomes often focus on the statistical classification of ‘hotspots’, often with the aim of characterising inequity. By comparison, the efficiency approach simply describes where events are located. However, it is worth noting that the polygons created within the STA could also be used for hotspot classification, for example in a manner similar to that of the popular approach SaTScan [18,19], which examines numerous spatial scan ‘windows’. An extensive literature exists that is devoted to overcoming the limitations of such windows as related to their shape (e.g., see Duczmal and Assuncao, 2017 [20]); given that the STA’s polygons are inherently flexible with regards to shape, an analysis of their use in classifying hotspots could usefully contribute to this literature.

Our findings expressly caution against the singular use of pre-defined administrative areas/boundaries (e.g., suburbs, local government areas, postcodes, ZIP codes) when delineating geographic TRs. Specifically, targeting limited intervention resources based on relatively small administrative areas (e.g., SA1s in Australia) may not be appropriate, since doing so might result in the delineation of numerous, discontiguous TRs, potentially precluding effective targeting of interventions. This is a manifestation of the small number problem. On the other hand, targeting resources based on relatively large administrative areas (e.g., SA2s in Australia) should only be undertaken with caution, since, as noted previously, the boundaries of such areas will not accurately reflect the spatial distribution of disease, except possibly by chance [7]. This is the impact of the MAUP. It is becoming gradually more widely recognised that data aggregated by political or administrative boundaries are unreliable for addressing many public health concerns, including the mapping of disease, and in fact might serve to undermine public health interests. In the US, for example, examination of blood lead levels in the population of Flint, Michigan, in 2015, using US ZIP codes (mean population size approximately 7500), delayed discovery of elevated blood lead levels in children of that county [21]. Thus, while targeting of interventions based on administrative boundaries is often expedient, for example due to funding commonly being distributed based on those boundaries, this reality should not preclude delineation of efficient TRs in the first place.

While we have developed the STA in the context of health, it could equally be ap-plied to guide geographic resource allocation and other interventions elsewhere. In criminology, for example, previous literature has recognised the importance of prioritising geographic TRs for distribution of limited policing resources in order to address crime (e.g., see Umar et al., 2020 [22]). In this context, the STA could be applied to datasets consisting of administrative units and the spatial locations of crime events, to produce appropriate TRs. Similarly, in transport planning, recent research has aimed to identify road section lengths with the greatest density of truck crashes, in order to guide policymakers in the allocation of highway patrol resources [23]. In this context, the STA could be applied to datasets consisting of lengths of road sections and the spatial locations of traffic incidents.

### Limitations

This study has several limitations. First, in developing the STA, we have not attempted to: (1) emulate Marxan + MinPatch’s attractive ability to produce multiple, different portfolios for a given dataset; (2) incorporate consideration of multiple different abundance features (another attractive ability of Marxan + MinPatch); or (3) derive estimates of precision around a given portfolio’s efficiency. These omissions were intentional, since our focus has been to demonstrate: (1) the use of both Marxan + MinPatch and the STA for delineating geographic TRs for healthcare interventions, and (2) that, for single outcomes (e.g., cases of disease or hospital admissions), such as are commonly examined in health, greater efficiency and control over logistical features is attainable using the STA. However, acknowledging that, in order to facilitate comparison and contrast of benefits and limitations associated with intervening in different areas, the availability of multiple different portfolios is often desirable in practice, including in health, we suggest that further work be undertaken to incorporate that functionality into the STA. With this in mind, it is worth noting that: (1) the STA’s unique separation of the process of polygon creation from the subsequent targeting of those polygons uniquely facilitates such an extension, and (2) if enough polygons were created, they could reasonably be sampled to produce precision estimates.

Second, the logistical factors we have considered do not constitute all possible logistical constraints that might impact upon the effectiveness of geographically targeted healthcare interventions in a given situation. Other potentially relevant factors include: politics, topography, military conflict, scarcity of resources and the availability of funding. Previous studies have also acknowledged the importance of such factors (e.g., see Lessler et al., 2018 [3]); however, it is outside of the scope of this study to incorporate them into the STA at this time.

And third, both the STA and Marxan + MinPatch are limited in that they cannot take into account any variation in the population density among geographic units in a region, or indeed any variation in the population sizes of those units. In conservation, the units used are typically geometric and uniform, e.g., raster cells or hexagons; however, in health, administrative units of varying shapes and complexities are typically used. This limitation is usually unavoidable. However, its impact can be mitigated by utilizing the smallest possible administrative units available, for example SA1s in Australia, as we have done in the stroke example.

## 5. Conclusions

Geographic targeting of finite healthcare resources and other interventions should be guided, in the first instance, or at least in conjunction with administrative boundaries, by sets of TRs that maximise efficiency while considering associated logistical factors. The boundaries of these TRs may or may not align with the boundaries of local administrative units. With comparison to Marxan + MinPatch, a method we have repurposed from conservation planning, we have shown how the STA, a new method we have introduced, can be used to efficiently delineate such regions. Given these findings, we suggest that application of the STA in both health and non-health contexts could help prevent inefficiencies arising due to allocation of resources guided purely by pre-defined administrative units.

## Figures and Tables

**Figure 1 ijerph-19-14721-f001:**
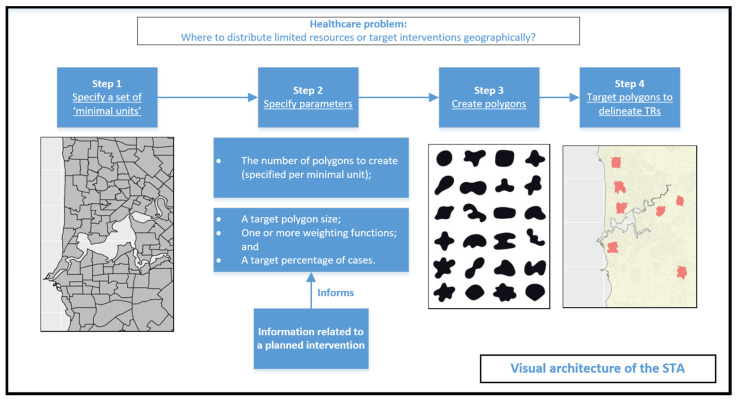
Visual depiction of the architecture of the STA in a healthcare setting.

**Figure 2 ijerph-19-14721-f002:**
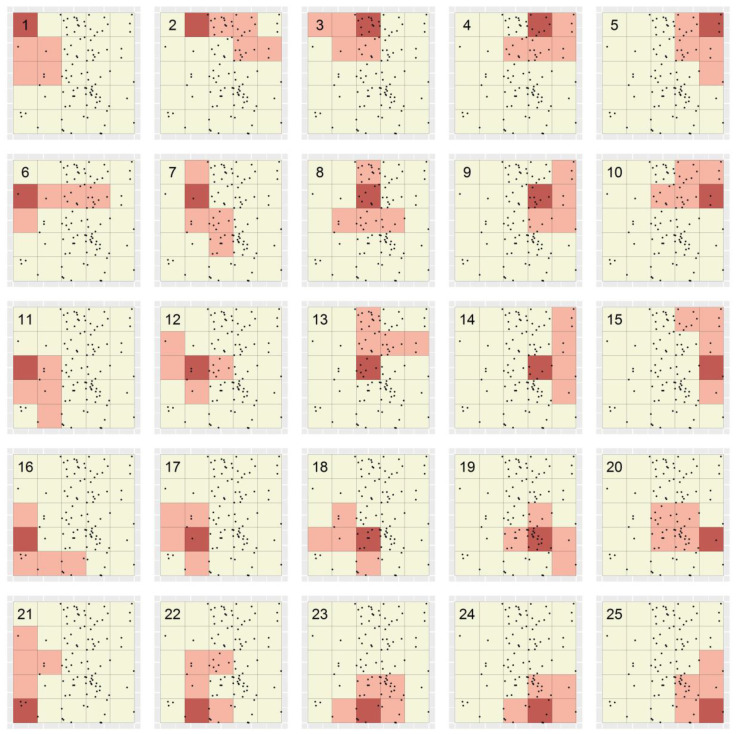
Polygons created within the STA applied to a simulated point location dataset of 100 disease cases. Black dots depict the locations of simulated cases, and grid lines represent the set of minimal units specified for the STA. In each panel, the pink shaded region represents the single polygon created beginning with seed unit (shaded red).

**Figure 3 ijerph-19-14721-f003:**
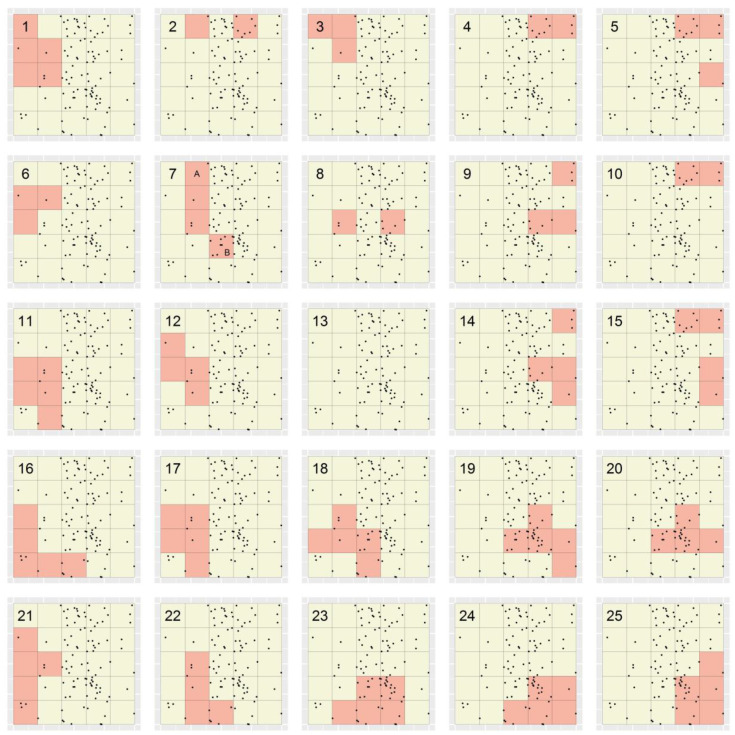
Reproduction of Figure 2 after removing overlap with polygon 13. As in Figure 2, black dots depict the locations of simulated cases and grid lines represent the set of minimal units specified for the STA. In each panel, the pink shaded regions represent the remainder of the respective polygons in Figure 2 after removing overlap with polygon 3.

**Figure 4 ijerph-19-14721-f004:**
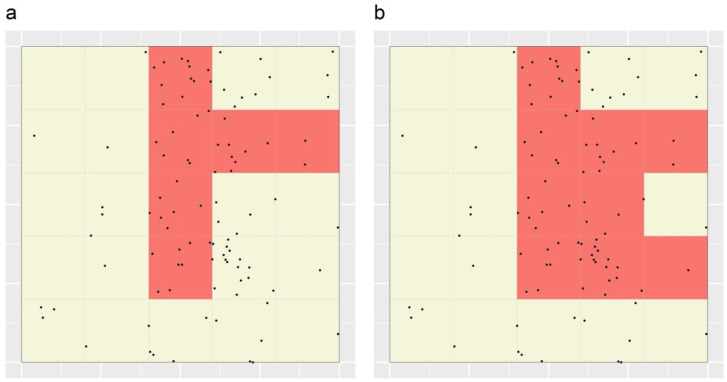
TRs delienated through applying the STA to the simulated dataset represented in Figure 2 and Figure 3, either (**a**) with splitting or (**b**) without splitting, based on a target case percentage of 40% of cases. Black dots depict the locations of simulated cases, grid lines represent the set of minimal units specified for the STA and the red-shaded regions represent TRs.

**Figure 5 ijerph-19-14721-f005:**
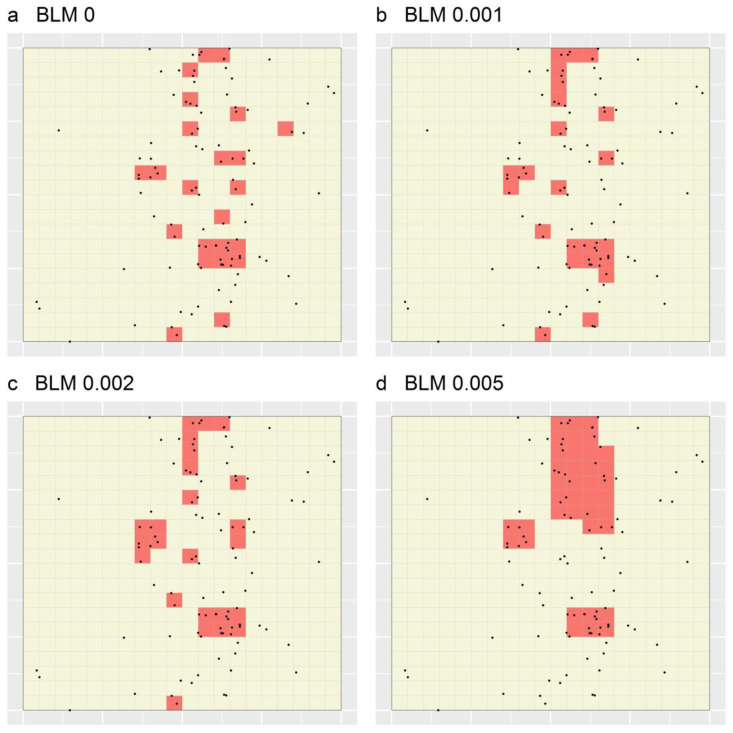
Optimal portfolios produced through applying Marxan to a simulated dataset while varying the specified BLM. Black dots depict the locations of the simulated cases, grid lines represent the set of spatial planning units specified for Marxan and the red-shaded regions represent the PAs delineated using Marxan.

**Figure 6 ijerph-19-14721-f006:**
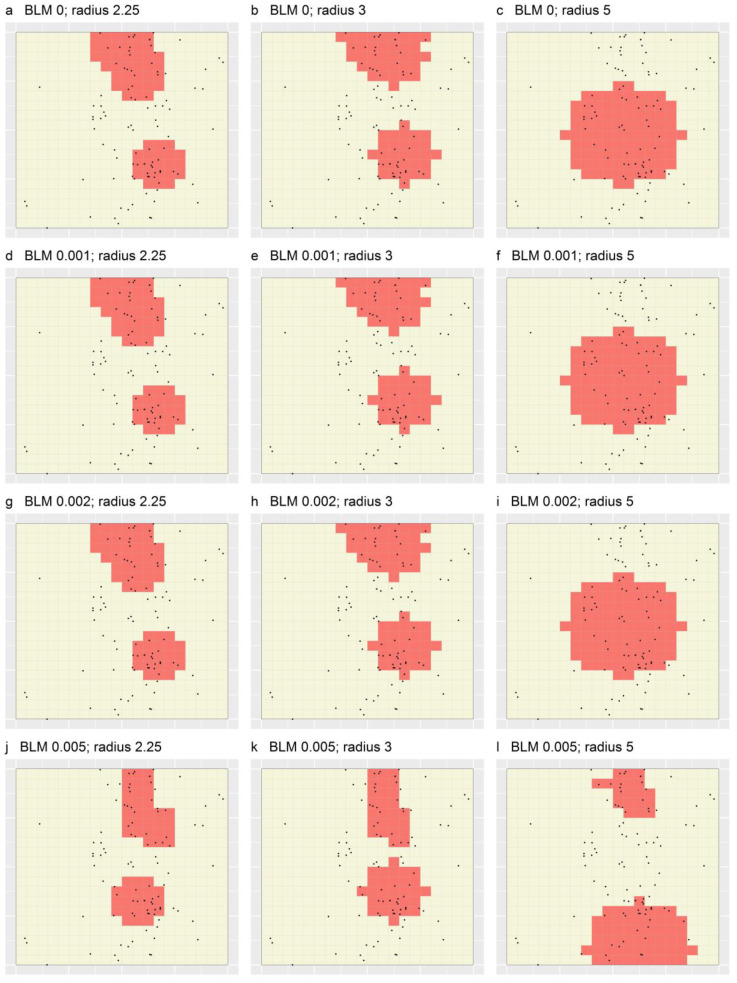
Optimal portfolios produced through applying Marxan + MinPatch to a simulated dataset based on 12 combinations of BLM and radius, and no simulated whittling. Black dots depict the locations of simulated cases, and grid lines represent the set of spatial planning units specified for Marxan + MinPatch. In each panel, shaded regions comprise ≥50% of cases.

**Figure 7 ijerph-19-14721-f007:**
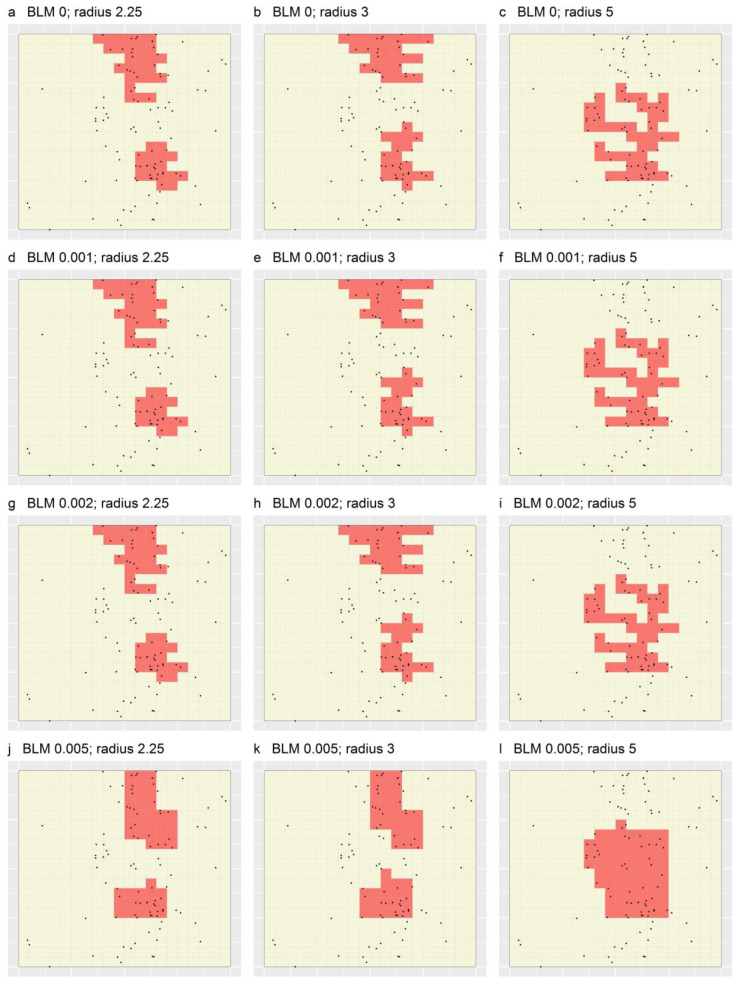
Optimal portfolios produced through applying Marxan + MinPatch to a simulated dataset based on 12 combinations of BLM and radius, and simulated whittling. Black dots depict the locations of simulated cases, and grid lines represent the set of spatial planning units. In each panel, shaded regions comprise ≥50% of cases.

**Figure 8 ijerph-19-14721-f008:**
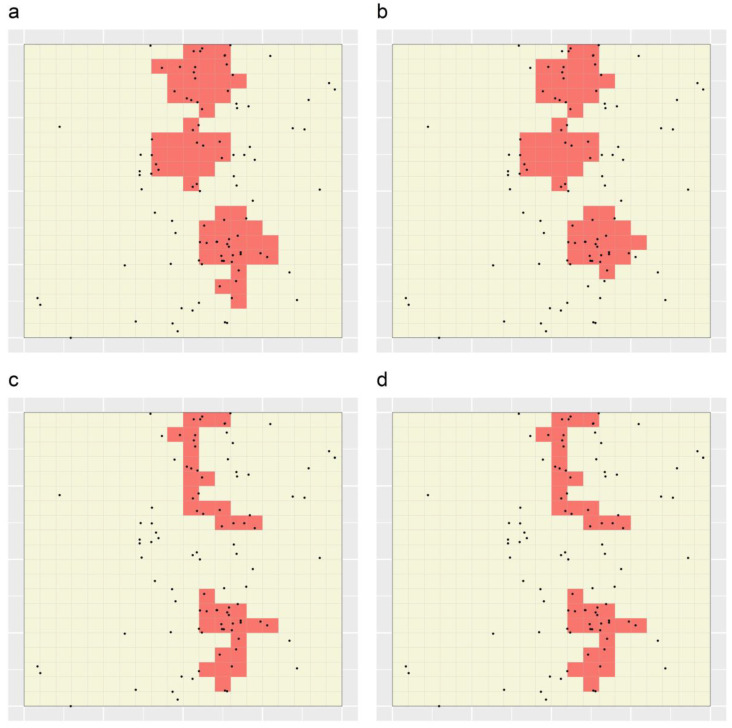
Sets of TRs delineated through applying the distance- and value-weighted STAs to a simulated dataset, either (**a**,**c**) with splitting, or (**b**,**d**) without splitting. Black dots depict the locations of simulated cases, and grid lines represent the set of minimal units. In each panel, shaded regions comprise ≥50% of cases.

**Figure 9 ijerph-19-14721-f009:**
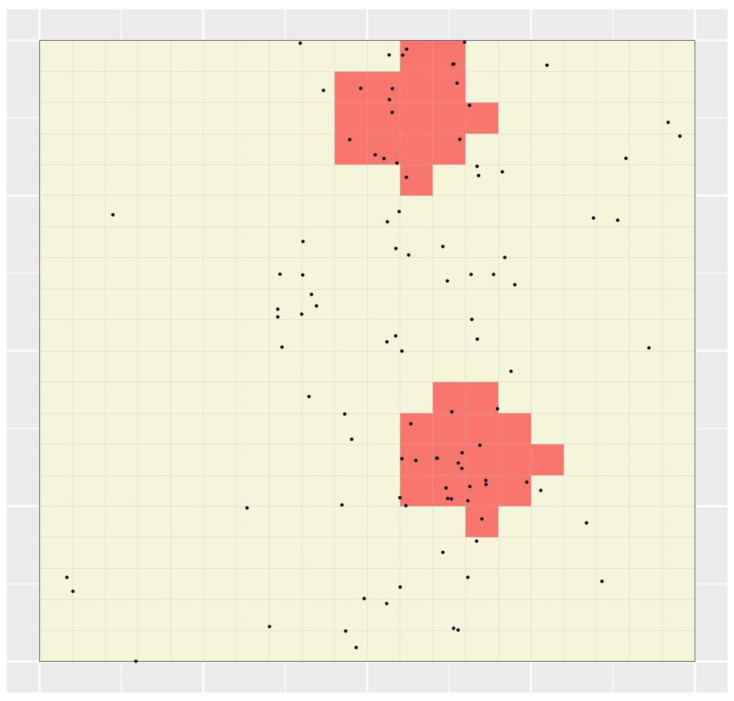
TRs delineated through applying a variation of the distance-weighted STA to a simulated dataset. Here, a requirement to delineate two, possibly contiguous TRs of a certain size has been stipulated. Black dots depict the locations of simulated cases, grid lines represent the set of minimal units specified for the STA and the red-shaded regions represent the TRs.

**Figure 10 ijerph-19-14721-f010:**
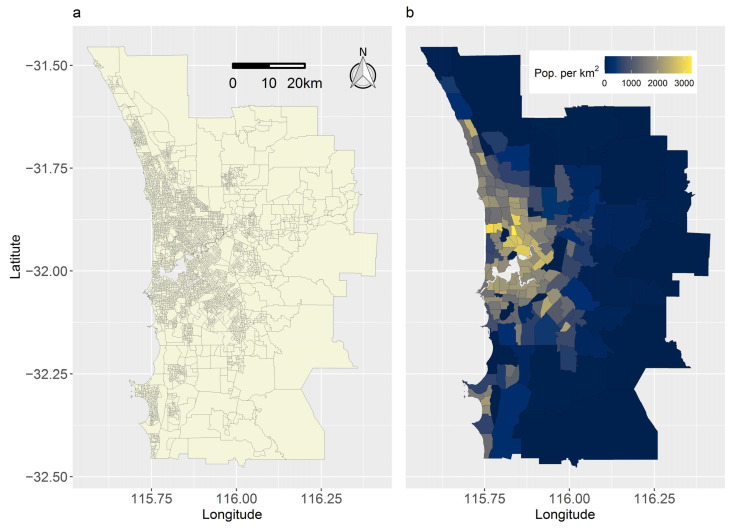
Administrative geography and population density of Perth in 2016. (**a**) SA1 and SA2 boundaries. (**b**) SA2-resolution population density.

**Figure 11 ijerph-19-14721-f011:**
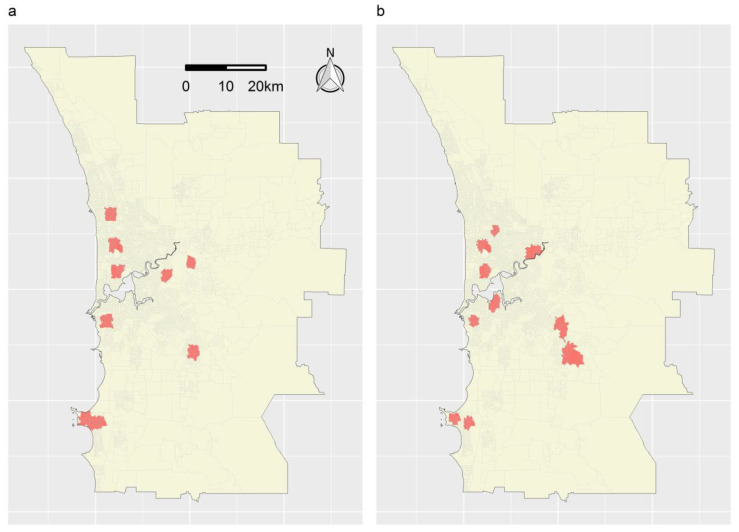
Sets of TRs comprising ≤ 15% of strokes in Perth in 2016. (**a**) TRs delineated using Marxan + MinPatch applied with a BLM of zero, a radius of 0.011 and simulated whittling. (**b**) TRs delineated using the distance-weighted STA applied without splitting. In each panel, grey lines represent the boundaries of SA1s and the red-shaded regions represent delineated TRs.

**Table 1 ijerph-19-14721-t001:** Efficiency data for the 24 optimal portfolios designated for sets of MinPatch portfolios produced through applying Marxan + MinPatch to a simulated dataset with a target case percentage of 50%.

		With Simulated Whittling		Without Simulated Whittling	
BLM	Radius	%Min. Units Targeted	Num. Target Regions	%Min. Units Targeted	Num. Target Regions
**0**	**2.25**	10.75	2	14.75	2
	**3**	11.25	2	16.25	2
	**5**	10.75	1	23	1
**0.001**	**2.25**	10.75	2	14.75	2
	**3**	11.25	2	16.25	2
	**5**	10.75	1	23	1
**0.002**	**2.25**	10.75	2	14.75	2
	**3**	11.25	2	16.25	2
	**5**	10.75	1	23	1
**0.005**	**2.25**	11.5	2	12.75	2
	**3**	11.75	2	13.75	2
	**5**	16	2	18.5	1

**Table 2 ijerph-19-14721-t002:** Exact efficiency data for portfolios produced through applying the distance- and value-weighted STAs to a simulated dataset. Values shown correspond to a target case percentage of 50%.

	With Splitting		Without Splitting	
STA	%Min. Units Targeted	Num. Target Regions	%Min. Units Targeted	Num. Target Regions
**Distance-weighted**	13.5	3	12	3
**Value-weighted**	8	2	8	2

## Data Availability

Restrictions apply to the availability of health data examined in this study. These data were obtained from the Department of Health, Western Australia and are available from that organization by application.

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
