# Peer review of "Improving the Efficiency of Geographic Target Regions for Healthcare Interventions"

_ijerph, 2022, doi:10.3390/ijerph192214721_

Round 1

Reviewer 1 Report

The authors have provided a new algorithm: the spatial targeting algorithm using simulated and real-world data.

  1. The analysis of results is very comprehensive and nicely written.
  2. The paper can be shortened substantially by removing some details. For eg: In section 3.4. Extended STA results can be removed and added to the supplementary materials.  The paper can be much shorter in length.
  3. Any plans to make the model streamlined for automated usage and ease of maintenance for the personnel with limited experience and knowledge. Any plans to handle this setback?  I mean calculating it for large-scale data is time-consuming, so are there any plans to speed it?
  4. Also is there any hypothesis that can explain why it either outperforms or remain competitive? and also, could you explain further how STA applications work outside of health as well? 
    Overall : accepted

       A nicely written paper that is much needed in the industry. Strong recommendation for acceptance.

Reviewer 2 Report

The paper explores the utility of Marxan + Minpatch methods over the proposed STA. A few specific comments are as follows:

1) Limited comparison between Spatial Targeting Algorithm (STA) and existing similar techniques in the literature. The efficiency of STA is measured in comparison to software tools, Marxan + Minpatch, etc.

2) The motivation of STA over Marxan + Minpatch is unclear. similarly, limitations in the software tools, Marxan + Minpatch are not addressed correctly. For example: in line no 70, Marxan + Minpatch considers important logistical factors; it is not clear which are those factors and in the proposed STA, which additional logistical factors are considered for the same purposes?

3) A detailed architecture framework helps to understand the flow of STA, which is missing.

4) Step 1 and step 2 of STA can be expressed with a visual running example; applying inverse distance-based weighting function, the calculation of new weight with a scattered set of points can be presented mathematically in equation(s).

5) In line no 191, 2.3 simulation study: here, the parameter setting for the simulation is not addressed.

6) Visualization of the value range with respect to each figure must be provided. For example, Figure 2 (overall) ranges between 0.0067 and 7.5 cases per minimal unit.

7) Other statistical factors, such as the population density of each region, radius selection, etc., how are they affecting the results?

Round 2

Reviewer 2 Report

Improved enough.